# A Brief on Nano-Based Hydrogen Energy Transition

**Rui F. M. Lobo** 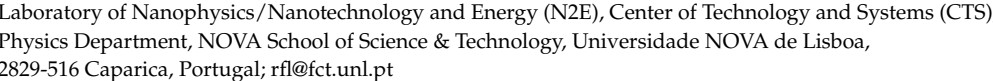

Laboratory of Nanophysics/Nanotechnology and Energy (N2E), Center of Technology and Systems (CTS), Physics Department, NOVA School of Science & Technology, Universidade NOVA de Lisboa, 2829-516 Caparica, Portugal; rfl@fct.unl.pt

**Abstract:** Considering the clean, renewable, and ecologically friendly characteristics of hydrogen gas, as well as its high energy density, hydrogen energy is thought to be the most potent contender to locally replace fossil fuels. The creation of a sustainable energy system is currently one of the critical industrial challenges, and electrocatalytic hydrogen evolution associated with appropriate safe storage techniques are key strategies to implement systems based on hydrogen technologies. The recent progress made possible through nanotechnology incorporation, either in terms of innovative methods of hydrogen storage or production methods, is a guarantee of future breakthroughs in energy sustainability. This manuscript addresses concisely and originally the importance of including nanotechnology in both green electroproduction of hydrogen and hydrogen storage in solid media. This work is mainly focused on these issues and eventually intends to change beliefs that hydrogen technologies are being imposed only for reasons of sustainability and not for the intrinsic value of the technology itself. Moreover, nanophysics and nano-engineering have the potential to significantly change the paradigm of conventional hydrogen technologies.

**Keywords:** hydrogen production; hydrogen storage; nanotechnology; nanocatalysts; electrolysis; green hydrogen; thermal desorption; hydrogen absorption

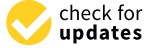



## 1. Introduction

The ongoing exploitation of fossil fuels as well as the current economic and geopolitical conditions have a significant impact on the availability of energy, which encourages the development of alternative fuels and environmentally friendly energy systems. Given this, hydrogen appears to be a viable alternative energy vector for many hybrid energy systems due to the high demand for the production of renewable energy [1].

Currently, there is no doubt that green hydrogen can play a very important role in energy transition; its challenges lie in improving its production, transport, and storage in conjunction with renewable production facilities. Increase in the hydrogen supply in some industries, namely in chemical and in all those that require strong thermal energy intensity, is particularly welcome. Consistent steps have been taken for its use in gas distribution networks and in sectors where electrification is more complex. The gradual achievements that are seen in electrical conversion technology (e.g., fuel cells) is a guarantee that mobility in its different aspects (road, sea, rail, aeronautics, and aerospace) is progressively resorting to the use of hydrogen. There already are some countries that have installed renewable hydrogen supply networks for mobility purposes [2].

Hydrogen costs and the increase in its introduction on the market are strongly dependent on its production efficiency and on its comparison with the corresponding evolution of competing fuels. Additionally, and according to the supply–demand law, the desirable existence of future available hydrogen surpluses will make this clean fuel the ideal solution for sustainability, meeting desired decarbonization targets. It should be remembered that, as with oil or natural gas, hydrogen does not exist naturally at our disposal without incurring extraction costs; energy, equipment, and operation financing are spent on this as well as with hydrogen. In addition, attention must be given to some relevant differences:

(i) hydrogen combustion does not emit greenhouse gases, in contrast to what occurs with fossil fuels; and (ii) the production of hydrogen uses abundant raw materials, whether water or natural gas. Bearing in mind that the Stone Age did not end for lack of stone, the "oil age" is not likely to come to an end for lack of oil. Indeed, this must constitute an inexhaustible stronghold for other industries, as in the case of plastics.

Fortunately, there have been important developments in scientific and technological research to boost the transition towards a hydrogen-based energy economy. This has more recently evolved not only through improving conventional production, conversion, and storage technologies but also by seeking new processes that lead to a significant lowering of inherent costs. Another important issue to be aware of (which has also seen significant improvements) is concerns with the safety of the technological processes applied.

The European Commission recently published the so-called Repower EU plan, which sets out the main objectives to gradually eliminate the EU's external dependence on fossil fuels before 2030. These include, among others, the increase in production and importation volumes of hydrogen, as well as the improvement of energy efficiency and the increase in renewable energies. It is envisaged that by 2030, around 50% of the hydrogen consumed in the EU will come from renewable sources, and additional funds will be made available to double the number of hydrogen vouchers and promote research and innovation in the field of renewable hydrogen.

The need for the drastic reduction of greenhouse gas emissions is a huge challenge facing humanity, which in the short term must be achieved using renewable gases and a more efficient use of the energy mix. For this, hydrogen assumes a predominant role, since its injection (up to 20%) into gas networks allows for most existing resources to undergo immediate decarbonization, with special relevance to the consumption of industrial resources [3].

It should also be noted that in this path towards effective decarbonization, hydrogen is associated with the desirable CCUS (Carbon Capture, Utilization, and Storage) and plays a significant role, as it mediates the conversion of greenhouse gases (e.g., carbon dioxide and methane) into synthetic liquid fuels [4].

The green hydrogen energy industry chain encompasses various stages, from renewable energy production to end-use applications, and nanotechnology plays a crucial role in advancing most of the relevant stages. In fact, nanotechnology innovations based on nanoscience drive advancements across the entire industry chain, and such developments enable hydrogen to be a viable and sustainable energy carrier, boosting the transition to a cleaner and more sustainable energy source in the future. The renewable energy process deals with the generation of renewable energy sources (solar, wind, geo-, hydro-) and nanotechnology leads to more efficient and cost-effective renewable energy harvesting technologies. The following two examples demonstrate this: one, quantum dots can enhance the efficiency of solar cells, while nanofluids can improve the cooling systems of wind turbines. Such renewable electricity is used to produce hydrogen through water-splitting processes (like electrolysis) or reforming of biomass; and two, nanoscience and nanotechnology have enabled the development of advanced catalysts, which are prone to enhancing the efficiency and lowering the energy requirements of hydrogen production methods.

Then, hydrogen needs to be stored efficiently to ensure its availability when required. Assembled nanostructures (such as carbon nanotubes) or hybrid supramolecular structures such as metal-organic frameworks have been explored as candidates for compact and high-capacity storage systems because they display high surface area and tunable properties. If hydrogen is not locally utilized, its transportation to end-users often involves pipelines or mobile means of transport. Nanotechnology can aid in developing coatings for pipelines that prevent hydrogen embrittlement and leakage and ensure a safe transmission. Finally, concerning the end-use of hydrogen, it is utilized in various applications, such as fuel cells for transportation, industrial processes, combustion, hydrogen propulsion, and power generation. Fuel cell technology has been developed after a period of stagnation mainly

thanks to nanotechnology by improving the performance and durability of the catalysts, which make fuel cells more cost-effective for widespread use.

Although at first glance the benefits of green hydrogen seem unrelated to nanotechnology, we must look deeper into the matter. Presently, for reducing carbon emissions and promoting an industrial sustainable transformation, green hydrogen is indispensable. In addition, the electroproduction of hydrogen using renewable energy is an effective way of raising energy efficiency, since it can work independently of the basic electric grid. Thus, this constitutes per se the first contribution to the increase in hydrogen production efficiency. Furthermore, when electroproduction performance is enhanced by catalyst effects acting at the nanoscale, a second contribution can be added to the increase in hydrogen production efficiency.

Advances already achieved with nanotechnology assume particular relevance and make it possible to foresee significant progress in most of the areas already discussed in which hydrogen is involved. Effectively, studies in energy and sustainability can make use of nanophysics and nanotechnology principles with the purpose of enabling the development and optimization of energy-related components and processes [5,6]. Nanotechnology for energy sustainability can be used in both bottom-up and top-down strategies and will be highlighted in this work—namely the most recent aspects involving carbon nanotechnology, fuel cells, and plasma catalysis. After stressing the importance and benefits of green hydrogen production, this work will emphasize the main methods and processes of hydrogen production and storage involving nanotechnology with a special focus on experimental research in the author's laboratory regarding these two subjects. Due to the pressing and complex nature of the subject, this brief manuscript utilizes general terms and is not written in accordance with the conventional scheme of a research work.

## 2. Green Hydrogen Benefits

Hydrogen is an important eco-friendly fuel alternative to conventional fossil fuels and is becoming more and more important for a sustainable society. Efforts towards extensive and effective decarbonization imply major changes in energy demands, especially for some productive sectors, in particular extractive and transforming industries. One can start by illustrating in more detail the beneficial role that hydrogen plays in processes where high energy intensity is required, as in the case of the aluminum industry, which is one of the most energy-intensive and $CO_2$-emissive industries (relying on electricity and fuel-burning) [7]. These emissions are mainly produced by the combustion of fossil fuels supplying the energy required for the various industrial processes. The fossil fuels currently used include natural gas, liquified petroleum gas, and light fuel oil. Hydrogen can be applied in internal combustion engines or in fuel cells to produce heat and electricity. In the latter case, it is a very efficient option due to its high electrochemical activity, while also accounting for a nearly 100% reduction in emissions. This refers to the premise that solid oxide fuel cells can be applied in a range from 1 kW up to 100 MW or even more. Their high-temperature operations (supplied for instance by a solar thermal source) enables the production of several grades of waste heat to be then recovered for process heating for power increase or even for the internal reform of hydrocarbon fuel to $H_2$ [8,9]. An $H_2$ SOFC can be used to produce high-grade heat for smelter and electricity, which returns to the grid, thus reducing the net electricity consumption of the system (the electricity required by the smelter is usually supplied by the grid). Typically, the SOFC operates with an electrical efficiency of at least 60% and a combined heat and power efficiency higher than 90% [10]. Advances in SOFCs have been pursued by several authors using nanoengineering to reach efficiencies of the order of 50% and significantly decrease the operating temperature. These efforts are focused on the development of ultra-thin films supported with metallic foils.

In a conventional hydrogen fuel cell, the cost of the electrodes could be considerably reduced by significantly decreasing both the amount and the durability of the platinum catalyst needed for the application. This cost is also directly related to the support and its influence on catalytic activity. The contribution of nanotechnology to induce a breakthrough

has been invaluable. For instance, using carbon nanotubes as catalyst support-based electrodes instead of carbon powder makes possible the reduction of the amount of Pt, which is used more than 2/3 times in comparison with the traditional PEMFC. Carbon nanotubes can make the fuel cell more stable [11] and have higher corrosion resistance performance. In addition, they can reduce the formation of surface oxides and the corrosion current.

There appears to be a clear link between the availability of energy and the state of the environment generally for energy-intensive businesses (such as aluminum, cement, and ceramics). Living Cycle Assessment studies can be used to compare the performance of various energy sources and are a well-established instrument for sustainable growth in the industrial sector [12,13]. One of the most promising methods to reduce the various emissions from the processes has been demonstrated to be the use of green $H_2$. Despite being created using the same energy source and taking into consideration emissions from the operation of the hydrogen technologies, green hydrogen is much more environmentally friendly than thermal energy-based electricity. The effectiveness of electrolysis and $H_2$ technologies differs. From a purely technical perspective, hydrogen technologies are viable for use in all energy applications and sectors, including electricity, heating, industry, and transport. Indicatively, a 12 MW hydrogen-fueled power plant in Italy has operated using a combined-cycle gas turbine and reformed $H_2$ with an estimated production of 60 million kWh a year [10]. In some industrial applications, brown hydrogen is prone to account for larger energy savings than green hydrogen, but the progressive efficiency improvements in the technologies of electrolyzers provide hope for a cleaner tendency based on the latter process. Promoting research and development in hydrogen production and utilization methodologies will certainly be the key to increasing environmental benefits. Figure 1 schematically displays the central role of hydrogen in the context of energy flux conversion. Hydrogen produced by renewable energy (green hydrogen) is the most environmentally beneficial option in terms of $CO_2$ equivalent mass. By enabling the local generation of thermal energy and electricity, hydrogen technologies also serve as a hub for decentralized green energy production. These technologies represent a reduction of more than 100% compared to conventionally used natural gas.

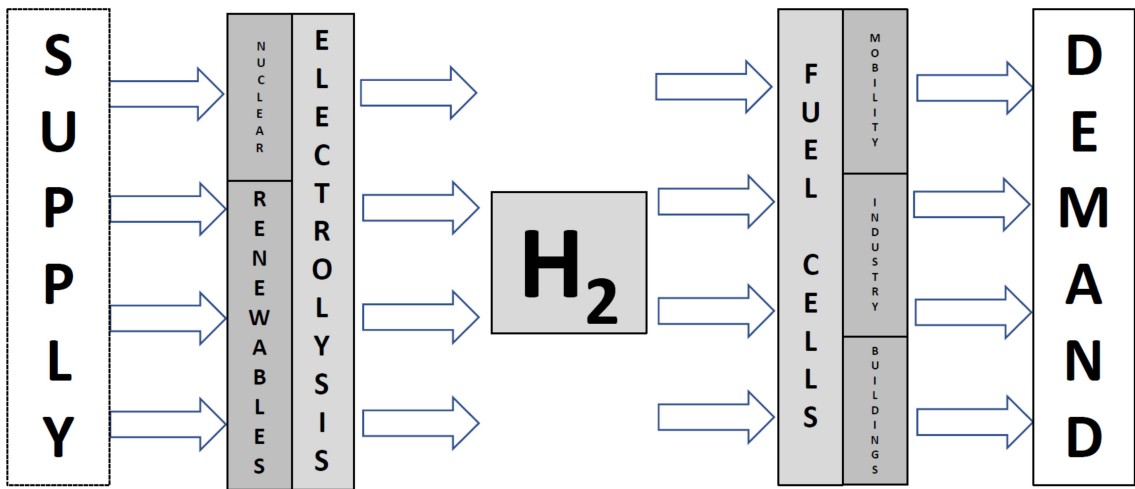

**Figure 1.** Clean Hydrogen Energy Flux Panorama.

## 3. Methods and Processes

These days, nanotechnology receives significant attention, which results in the raising of expectations among investors, governments, and businesses in addition to the academic community. Its singular capacity to create new structures at the atomic level has already led to the creation of innovative materials and gadgets with a wide range of potential uses. Among them, significant breakthroughs are especially required in the energy sector that

will allow us to maintain our increasing energy consumption, which increases both with the ever-growing population and with demand per capita. Solar, hydrogen, new-generation batteries, and supercapacitors are probably the most significant examples of the contributions of nanotechnology in the energy sector. Some significant contributions that have the potential for finding a solution to one of the great challenges of our time can be presented here, i.e., the production and use of green energy-based hydrogen without compromising our environment and from one exciting and multidisciplinary field, nanotechnology. Green hydrogen directly benefits from nanotechnological advances made in renewables (e.g., solar energy harvesting or wind turbines) and electrolyzers or fuel cells as well. Both contributions are important, especially reckoning that the energetic efficiency improvement of electrolyzers leads to the decisive saving of territory, whose occupation is usually a priority for other human economic exploration.

From the source to the consumer, hydrogen itself may carry and store energy. To be transported effectively, the renewable energy sources found in nature need to be primarily converted into electricity. It seems unlikely that hydrogen would need to be produced, but, fortunately, it can be accomplished using renewable energy sources and is easily turned into power utilizing fuel cell technology. Because of this, hydrogen is seen as an energetic vector preparing the foundation of an independent energy system. As a result, direct synthesis of hydrogen from renewable sources is anticipated in the future, negating all electrical, thermal, and mechanical losses.

(A)  Green Hydrogen Production

   (i)    Photocatalytic Water Splitting

With the ability to alter the energy bandgap, the addition of nanostructured components to PV cells increases flexibility, improves the effective optical path, and sharply lowers the likelihood of charge recombination. Nanocrystal quantum dots, which are nanoparticles typically made of direct bandgap semiconductors, have lately advanced PV technology and enabled the development of thin film solar cells based on silicon or conductive transparent oxide substrates with a nanocrystal coating. Due to their ability to emit multiple electrons per solar incident photon and the fact that they have distinct absorption and emission spectra depending on the particle size and shape, quantum dots and quantum wires are effective light emitters that can increase efficiency by adjusting to the incoming solar radiation wavelength.

In turn, PV energy can be used to break water molecules into $H_2$ and $O_2$ via photocatalytic electrolysis, which enables solar energy to be directly stored in the form of hydrogen. This photocatalytic process of water molecular dissociation is another type of splitting process (at temperatures under 1000 °C) such as electrolysis, thermochemical cycles, or thermolysis on defective carbon structures. For this purpose, a variety of semiconductor catalyst nanoparticle systems based on SiC, CdS, $CuInSe_2$, or $TiO_2$ can be used despite not yet being cost-effective due to low conversion efficiencies. Indeed, under photonic irradiation, nanoparticles (e.g., of semiconductor metal oxide on the order of 30 nm) sandwiched between transparent and conductive polymeric layers, can release and conduct electrons through the polymer before electron-hole recombination occurs. Figure 2 displays the scheme of a solar water splitting system using a composite semiconductor electrode, in which the space surrounding the nanoparticles is filled by an aqueous electrolyte. Regarding the example of $TiO_2$, its bandgap is 3.2 eV, and only UV light can be utilized for this purpose.

The good performance of the photocatalyst cells is, however, limited by the recombination velocity of the photo-generated electron/hole pairs, which is in general fast, lowering in this way the conversion efficiency. Improving research can make use of specially designed organometallic complexes. For example, the following occurs in a complex with a central rhodium atom and two peripheral ruthenium atoms (each one in opposite locations): when these latter receive a photon, the rhodium transfers an electron to water, giving rise to the splitting in $H_2$ and $O_2$. Nano-photocatalysis has also been extended to the high temperature-splitting of methane into hydrogen and carbon (coined by artificial

photosynthesis). At any rate, it is worth mentioning that despite nano-photocatalysis-based devices representing an attractive option for direct hydrogen generation from a primary renewable energy source, they are mainly interesting for small-scale or local applications and not so much for large-scale hydrogen production, since the processes are not easily scalable.

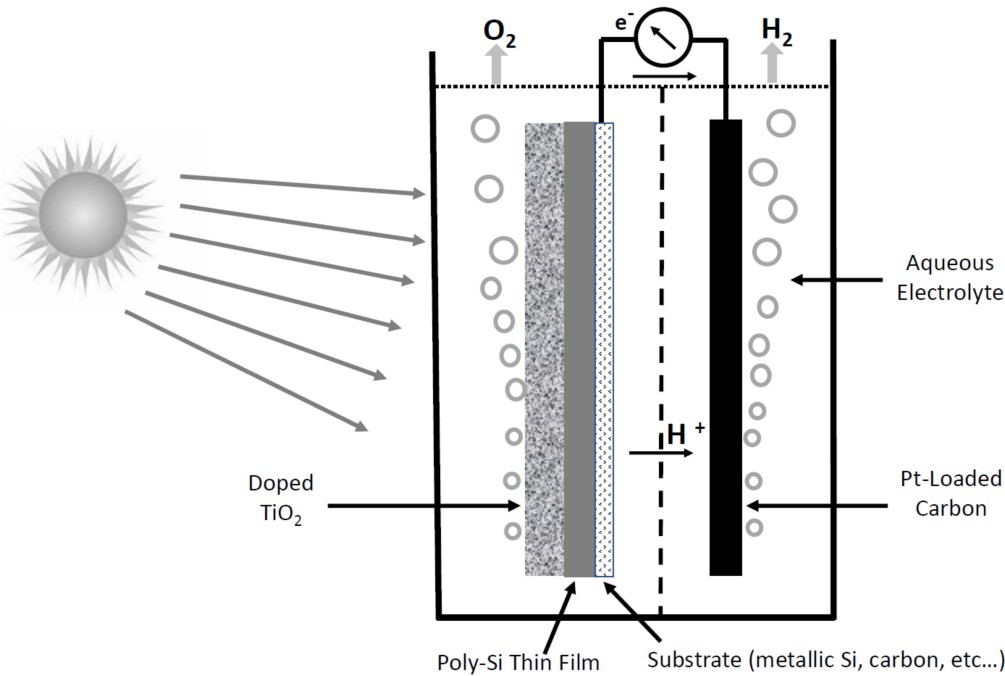

**Figure 2.** Solar water splitting system using a composite semiconductor electrode.

(ii) Solar-hybrid Electrolysis of Water

The coupling of water electrolyzers powered by photovoltaic panels (or by wind turbines) is a better scalable alternative, although mostly limited by the efficiency of the PV panels (or wind generators) and mismatch between the panels (aeolian turbines) and the electrolyzer. A renewable generator must operate at its maximum power point to transfer electricity to the electrolyzer as efficiently as possible (MPP). Temperatures of the generator and electrolyte have an impact here. The operating point of the entire system should be equal to the MPP of the solar generator for a PV-electrolysis system to be able to produce hydrogen efficiently. This is often accomplished through DC/DC converters, which match the electrolyzer's input to the solar generator's output. Another option is to change how the solar generator or the electrolyzer is configured [14].

Therefore, water electrolyzers powered by renewable sources display net efficiencies that are dependent on one hand on renewable source efficiency and on the other hand on the efficiency of the electrolyzer. This offers promising perspectives because nanotechnology advances have been used to improve both of the independent technologies on their own. Various examples confirm significant improvements in the performance of wind turbines via the incorporation of composite nanomaterials in the manufacture of windmill blades (e.g., carbon nanotube-filled epoxies), so the resulting longer blades increase the amount of electricity generated. Perovskite solar cells have also been used to make some significant advances in the efficiency of PV panels. Since 2009, when reports of their efficiency were said to be at 3%, they have achieved incredible success, quickly increasing it to over 25% now. Perovskite solar cells have rapidly increased in efficiency, but there are still obstacles to overcome before they can be considered a viable commercial technology. The Shockley–Queisser limit can be overridden by using multi-junction solar cells in experiments, allowing for the absorption and conversion of photons throughout a wider wavelength range without increasing the loss of thermalization [15]. Once experimentally

ensured performances reach the theoretically imposed efficiency limits of renewables, it is obvious that the next disruptive step in the production of hydrogen is to be able to significantly increase the efficiency of current electrolyzers. Also, thanks to the unprecedented control over the size, structure, and organization of matter, many nano-scientists and nanoengineers obtain novel materials or novel device designs with unique physical or chemical properties, which already contribute to the overcoming of some challenges. In general, the efficiency of water electrolysis can be achieved by increasing the surface area for reaction and reducing the energy required to split water molecules.

(iii)  Plasma Catalysis

Furthermore, a viable alternative to the use of electrolysis has been based on the progress made with plasmas and plasma catalysis. Hydrogen production via the plasmolysis of water has been studied from both theoretical and experimental perspectives [16–24]. Key kinetic processes driving the breakdown have been identified, and Rehman et al. detailed a kinetic modeling of hydrogen production from water vapor plasmolysis in plasma microreactors [22]. High electric fields at a relatively low voltage can be used in plasma microreactors to generate energetic radicals like H, OH, and $HO_2$, among others, at an atmospheric pressure. By a sequence of events, these active species combine to eventually generate $H_2$ and $O_2$. Corona discharges and dielectric barrier discharges are two types of nonthermal atmospheric pressure plasmas (DBD). Corona discharges generate a high concentration of radicals, but they have a very low power, which is inconvenient for water vapor splitting. To enhance power, voltages could be raised; however, doing so causes the corona to transform into arcs. Arcing could be avoided by establishing corona discharge in a pulse periodic mode or by covering one or both electrodes with a dielectric material. The discharge current may be controlled, arcing between two electrodes can be prevented, and homogenous discharge can be produced by adding a dielectric layer [25]. To overcome these challenges and benefit from both the high radical concentration of corona discharges and the homogeneity of plasma discharge at a higher power in DBD, various DBD-corona hybrid reactors have been developed [26]. In addition, water vapor plasma can be characterized using optical emission spectroscopy, which can confirm the existence of nonlocalized thermal equilibrium in plasma bulk via the sequential order of the characteristic temperatures ($T_e > T_{rot} > T_{exc}$). A typical experimental schema is displayed in Figure 3.

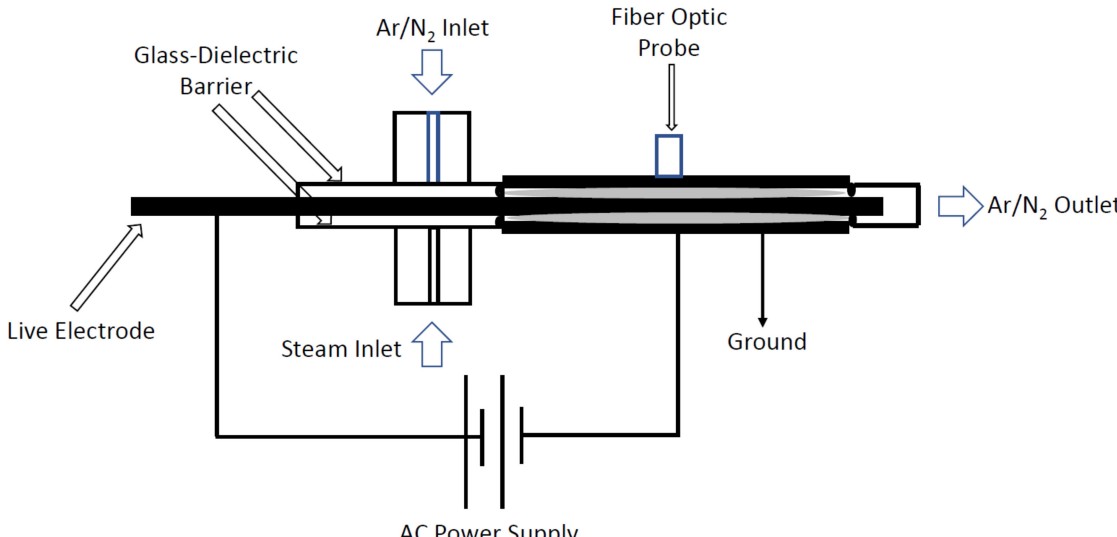

**Figure 3.** A simple DBD-corona reactor.

These types of experiments have shown the promise that the production rate of hydrogen, as well as its predicted costs, are competitive to the conventional electrolysis process, with the benefit of reduced equipment size and low power consumption [27]. From

a general point of view, the control of plasma processing methodologies must be preceded by a clear understanding of the physical and chemical properties of plasmas [28].

A non-equilibrium plasma (also called nonthermal or cold) is a plasma that is not in thermodynamic equilibrium because the electron temperature is much hotter than the temperature of heavy species (ions and neutrals). As only electrons are thermalized, their Maxwell–Boltzmann velocity distribution is very different from ion velocity distribution. At temperatures that are attained in plasmas, it is necessary to consider the population of excited electronic states, and their density becomes very great as the ionization energy is approached. This means that extremely high temperatures are required to produce a significant ionization of a gas under equilibrium conditions. When total energy input is considered, non-equilibrium plasmas have a great efficiency in using energy, as one may choose conditions so that electrons have a very high probability for a given process, and at the same time, no energy is wasted in heating the gas and container. The existence of significant populations of energetic electrons (i.e., 10 eV, or higher) allows the efficient, non-thermal dissociation of molecular gases to produce high concentrations of reactive radical species. For synthesis, non-equilibrium plasmas are preferable because they could be able to open chemical pathways that thermal methods would not be able to. Because plasma produces kinetic energy in the form of energetic electrons, which can contact gas molecules to activate them, it is undoubtedly a viable medium for energy efficient chemical conversion.

Such collision events lead to dissociation, ionization, and excitation, generating reactive species (i.e., radicals, ions, excited species) that can easily form new products.

The steady state of a plasma can be disturbed by the introduction of a solid surface in a way that can result in a catalytic action leading to a higher degree of chemical reactivity of the system under study. An example of that is the synthesis of methanol from $CH_4$ and $CO_2$ in a DBD, where it was found that the process was enhanced with the introduction of a catalyst [29]. Nowadays, plasma-assisted catalysis has several applications in environmental clean-up, removing common pollutants, and in the directed synthesis of added value products such as in the reforming of hydrocarbons into fuels [30]. This can reduce commonly occurring problems of catalyst stability such as sintering at high temperatures or poisoning by species such as sulfur. Certain advantages such as low-temperature operation, high selectivity, and improved energy efficiency are priceless. It is timely to remember that energy efficiency means using less energy to carry out the same work—and in the process, cutting energy bills, reducing pollution, and combating climate change. This way, energy efficiency reduces energy waste and lowers energy costs.

The mechanism of plasma catalysis is complex and far from completely understood. In thermal catalysis, the catalyst is activated by heat, whereas in plasma activation, the energy comes from an electrical discharge. Ions, reactive atoms, radicals, excited species (electronic and vibrational), and photons are all products of electron–gas collisions. A non-thermal plasma has high-energy electrons out of equilibrium but no gas heating. Particularly at atmospheric pressure, when quenching, recombination, and neutralization are quick processes, many plasma-created species are short-lived. Vibrationally excited species interacting with catalytic surfaces may also play a role. The interactions in plasma catalysis are either from the plasma within the catalyst or the catalyst affecting the discharge. On the one hand, examples of the effect of the plasma on the catalyst are the formation of radicals and excited states, reduction in active metal, and modification of the properties of the catalyst; on the other hand, the effects of the catalyst in the plasma are adsorption on the catalyst surface, influence on plasma generation, and a packed-bed effect. All these synergistic effects converge to enhance energy efficiency, improve selectivity, increase the concentration of the active species, and improve catalyst activity and durability. Plasma interactions with ions, electrons, or photons can alter surface characteristics in addition to producing reactive species above the catalyst surface. By adding catalytic materials, the discharge may change in nature, such as from filamentary micro-discharges to surface discharges, or by changing the dielectric effects that affect it. While cold plasma is only

employed periodically for a short portion of the time needed to saturate the adsorbent catalytic material, plasma catalysis frequently offers advantages over a continuous heat system. This results in a large energy savings.

Low-temperature plasma reduces NiO to Ni during the processing of a NiO-Al$_2$O$_3$ catalyst with atmospheric pressure methane plasma [31]. This reduction is complete when no more CO$_2$ is evolved. Thermally, reduction occurs at temperatures of more than 400 °C; however, in this instance, reduction occurs at lower temperatures. The subsequent Ni-catalyzed process, CH$_4 \rightarrow$ C + 2H$_2$, which uses the fragmentation of adsorbed CH$_4$ on active sites of the catalyst surface to create active adsorbed carbon and hydrogen, produces hydrogen with a high degree of selectivity. Carbon sometimes manifests as nanofibers; a Ni-catalyzed process is typically feasible at temperatures above 600 °C, demonstrating increased energy efficiency for low-temperature plasma catalysis over conventional thermal processing and demonstrating a synergistic effect for CH$_4$ decomposition, where both plasma and catalyst are essential. Most nanocatalysts are in the form of nanoparticles and when their diameters are smaller than about 5 nm, they can become powerful catalysts, as in the case of gold nanoparticles for the oxidation of carbon monoxide. The reactivity of the nanoparticles depends not only on their size but also on the material on which they are supported. However, it has been found that the dominant effect is that of the gold nanoparticle size, with the nature of the support playing a secondary role [32]. The stability of hollow nano-cages like metal-fullerenes or metal-organic frameworks can be sustained under non-thermal plasma activation and in the presence of water.

To reduce the effect of CO$_2$ on global warming, dry reforming of methane for syngas production is a convenient technique. Since CO$_2$ has high thermodynamic stability, a large amount of energy is needed to activate CO$_2$ gas. For a thermal activation, typically reaching 1500 K, a substantial amount of energy is lost in heating the entirety of the gas. The sole feasible method of thermal CO$_2$ conversion is the dry reforming of methane, which is the simultaneous conversion of CO$_2$ and CH$_4$ to produce syngas (CH$_4$ + CO$_2 \rightarrow$ 2CO + 2H$_2$). Thermal-catalytic CO$_2$ splitting is also very energy intensive. According to Le Châtelier's principle, a reaction will move backward if the reactants' pressure is increased; hence, it is typically conducted at an atmospheric pressure. However, this process is also endothermic, with a typical reaction enthalpy of 2.56 eV per converted molecule, like direct CO$_2$ splitting. As a result, it must be conducted using a catalyst at high temperatures (600–900 °C) (Ni, Co, noble metals, Mo$_2$C). This inefficiency conversion can be avoided by using non-thermal plasmas, which, in addition, are an attractive alternative to the conventional (catalytic) thermal route. This makes it possible to carry out chemical reactions at temperatures near ambient temperatures in a reactor that does not require much energy. Although the overall gas kinetic temperature is low in a non-thermal plasma, electrons are accelerated by the applied electric field to energies between 1 and 10 eV, which are sufficient to dissociate most chemical bonds (notice that the standard reaction enthalpy for CO$_2$ dissociation is 2.9 eV). Thermal plasmas are ineffective for the efficient conversion of CO$_2$ due to their nature. The highest energy efficiency is limited to the thermodynamic equilibrium efficiency and corresponding conversions of 47% and 80% at 3500 K, respectively, because ionization and chemical reactions in thermal plasmas are temperature dependent. In contrast, lab-scale efficiencies of up to 90% have already been observed for non-thermal plasmas [33]. In non-thermal plasmas, the electric field provides electrons with their energy. Via collisions, this energy is then dispersed across various channels of excitation, ionization, and dissociation as well as elastic energy losses. Besides CO$_2$ conversion, other processes display synergistic effects when plasma and catalytic effects are combined, for instance plasma-enhanced nanoparticle-catalyzed CNT growth and natural gas reforming (steam reforming), catalytic synthesis of inorganic nanowires, and synthesis of large-area graphene films. Hydrocarbon reforming of natural gas has received particular attention because it is a major source of hydrogen gas, which in turn is the current focus of the world due to its potential to be a clean energy vector for replacing fossil fuels and implementing a sustainable hydrogen-based economy. There are unambiguous warnings that current

world energy trends are not sustainable, and hydrogen is a product that has the potential for being an emission-free alternative fuel. Currently, almost 90% of $H_2$ is produced via high-temperature steam reforming of natural gas or light oil fraction. However, this process where water vapor reacts with methane still suffers from some inconveniences: (1) moderate thermodynamic efficiency; (2) fouling/coking on the catalyst surface, sulfur impurities with eventual deactivation, and high-temperature requirements of the reforming reaction (a condition that reduces the energy efficiency of the process); (3) need for further separation of hydrogen from the syngas product or following a water shift gas reaction that results in carbon dioxide. The process is endothermic (165 kJ/mol), making high energy requirements, so the wet reforming of methane is a less favorable route. Thus, plasma reforming may offer several advantages. It was found that the effect of steam addition increased $CO_2$ yield, despite a decrease in methane conversion [34]. As a result, the plasma-assisted catalytic reforming of natural gas in a DBD reactor came to offer significant advantages, including operation at a lower operating temperature, selectivity for the creation of products with value added, and a very quick switch-on time. Operation at a lower temperature results in less coke production and catalyst poisoning, which increases the lifetime of the catalyst. In any industrial context, this is a crucial variable that partially offsets capital expenses. The plasma-catalyst synergy effect may be attributed to the presence of vibrationally excited $CH_4$ molecules produced in the plasma [35]. In this method, the selective generation of syngas and perhaps other value-added fuels like methanol or formaldehyde may be made possible by the plasma-catalytic reformation of natural gas.

As a result, the use of intermittent excess energy (such as that from renewable energy sources) to store this excess electrical energy in the form of liquid fuels may eventually be possible thanks to the quick plasma switch-on time (or hydrogen storage). Large-scale gas conversion requires a significant amount of energy, making the process of energy efficiency crucial. Thus, plasma-catalytic reforming must drastically reduce energy costs as much as possible. Plasma catalysis is carried out through very fast reactions, with minimal waste production compared to chemical synthesis. Plasmas also have advantages in terms of the speed of preparation of the catalysts involved, low energy requirements, the production of widely distributed active species, as well as increased selectivity and an adequate catalyst lifetime. Moreover, the catalyst can be activated via plasma processing both during the catalyst synthesis phase and during the catalyst pre-treatment phase. When the size of the nanoparticles is a function of the intensity of the discharge, it is possible to use the discharge to lower pre-treatment temperatures while still producing smaller nanoparticles with a significant improvement in dispersion.

(B) Improving Hydrogen Storage

Finding a safe and not costly technology that can expend as little energy as possible for hydrogen storage and delivery is an issue that presently involves many research laboratories around the world. Despite being established technologies for industrial applications, liquefied and compressed hydrogen require energy consumption and adequate steps to address problems at high pressure (up to 100 MPa) and low temperature (around 20 K). Instead of requiring high hydrogen pressure and cryogenic temperatures, storing hydrogen in suitable materials can result in a more portable and secure method. The utilization of hydrogen storage solids as active components and for use in fuel cells has been the subject of numerous studies. Presently, one of the most effective methods for storing light and small hydrogen molecules is through hydrogen adsorption, either chemisorption or physisorption. Chemisorption techniques have the drawback of binding them too tightly, and the storage system must operate above room temperatures (>400 K) for discharge while recharge is highly exothermic. On the other hand, with physisorption, significant adsorption is usually achieved at temperatures below 100 K, and typical values obtained are about 50 g$H_2$/kg of activated carbon at −190 °C under a pressure of 6 MPa. By shrinking the adsorbing material to a nano-size range in both situations, hydrogen adsorption kinetics can be enhanced [36]. Mesoporous materials have high surface areas (usually greater than 1000 m$^2$/g) and regulated pore size and shape. As a result, nanomaterials that satisfy the

criteria of high surface area, tailored pore size and shape, high storage capacity, controlled desorption, and safety can be used to produce high hydrogen adsorption capabilities. Many high-surface nanoporous materials have been studied primarily in groups based on simple elements (nanocarbons, nanoborons) and metal-organic frameworks [37]. This nanotech improvement follows decades of research by several authors on activated carbons, where just a small portion of the pores are small enough to interact significantly with hydrogen molecules in the gas phase. Therefore, activated carbons have been useless as hydrogen storage materials. The physisorption of hydrogen on activated carbon with a regulated pore size distribution and the right surface chemistry are novel possibilities to efficiently store hydrogen. Other carbon nanostructures [38,39] such as carbon nanotubes, graphite nanofibers, and graphite nanoparticles are additional porous substitutes for activated carbon. Carbon nanotubes (both MWCNT and SWCNT) are suitable candidates for hydrogen storage [40] and different mechanisms of both physisorption and chemisorption of hydrogen on nanotubes have been suggested to describe these processes [41]. The maximal degree of SWNT hydrogenation depends on the nanotube diameter, and for diameter values around 2.0 nm, nanotube–hydrogen complexes with close to 100% hydrogenations exist and are stable at room temperature [42]. This corresponds to about 10 wt% of the hydrogen storage capacity of the outer SWNT surface. Also, it is worth noting that nanomaterials with high hydrogen storage capacities have been obtained through the modification of highly porous carbon nanofibers (surface area of 2000 $m^2/g$) with Ni nanoparticles [43]. Carbon nanotubes combine the advantages of low density with high mechanical resistance, chemical inertia, and compactness. Most of the adsorbed hydrogen is released at room conditions of pressure and temperature.

Considering both safety and cost, metal alloys ($ZrV_2$, $Mg_2Ni$, PdRh, FeTi, $LaNi_5$) or metal hydrides ($PdH_x$, $MgH_2$) display good performances for hydrogen storage in comparison with conventional methods (cryogenic liquid or compressed gas). These systems achieve a storage capacity comparable to that of liquid hydrogen or pressurized hydrogen, depending on the mass of the metal or alloy, because hydrogen atoms are inside the lattice of the metals or alloys. Before molecular hydrogen is released again after additional heating, the metal divides the $H_2$ into H atoms. $MgH_2$ is a high-temperature hydride that stores 0.07 kg of hydrogen per kilogram of metal, according to gravimetric analysis [44].

The ability of metal/alloy hydrides to store hydrogen, the number of reversible storage cycles, and the kinetics of hydrogen absorption/desorption are three crucial aspects to consider. Nanopowder particles also have some obvious benefits because they are directly related to the constitution of alloy microstructures. Mg-Ni alloys are inexpensive, lightweight materials that allow for effective hydrogen absorption without the need for a protracted activation process. Moreover, by doping metal hydrides with nanoparticles, like in the instance of $NaAlH_4$ doped with titanium nanoparticles, capacity can be increased. According to reports, fluorographene nanosheets (FG) improved the capacity of magnesium hydride to absorb and release hydrogen. The $MgH_2$-FG composite's de-/rehydrogenation characteristics were improved because the fluorographene nanosheets in it servee as hydrogen transfer centers and sped up hydrogen incorporation and dissociation [45]. Generally, metal hydrides have the disadvantage of having a weight greater than the equivalent amount of liquid hydrogen, but they represent an enormous saving in leakage and the energy charge needed.

By putting the metal on the surface of porous materials after such materials have been manufactured, one can use porous supports to preserve the small size of metal nanoparticles. By functionalizing Pd nanoparticles with tri-alkoxysilane terminal groups, a different method of obtaining highly dispersed nanoparticles for improving the hydrogen adsorption performance of silica-supported palladium can be used. These functionalized nanoparticles are incorporated into the porous structure of silica supports by copolymerization with tetra-alkoxysilanes in the presence of a surfactant [46].

Surface- or interface-related sites take on critical importance in tiny systems and can alter the overall solubility of hydrogen. Hence, a lot of focus has been placed on thermal desorption research and calibration to improve the kinetic properties of hydrogen storage in solid media. In contrast to the determination of static sorption isotherms, thermal desorption spectrometry ensures fair accuracy in monitoring hydrogen absorbed in solid materials and is ideal for monitoring hydrogen evolution after the application of a thermal ramp [47]. Identification of developed gas species is made possible via mass spectrometry, where data on desorption temperatures provide information on the binding energy of adsorbed molecules, which changes depending on the type of adsorbate/surface material. Finding carbon nanomaterial assembly-based hydrogen active storage media with a low weight and good performance is another objective. In Table 1, the relative performance of a few carbons' nanomaterial assembly-based electrodes was chosen for comparison. The recently developed molecular-beam thermal desorption spectrometry (MB-TDS) method has been used for this purpose [48,49]. With some benefits, this method was created to detect the hydrogen released by the smallest amounts of solid samples [48,50]. The accurate assessment of the hydrogen mass absorbed on a solid sample has significantly improved thanks to MB-TDS (note that this novel thermal desorption variation is based on an effusive molecular beam [49]). There is a 20% improvement in the signal-to-noise ratio for trace hydrogen, and prior calibration using a chemical standard is not necessary. Table 1 displays some typical experimental values for hydrogen uptake in several porous solids at normal pressure using MB-TDS.

**Table 1.** Hydrogen uptake typical experimental values for different porous solids at normal pressure.

| Material | Temperature | Loading |
| --- | --- | --- |
| Activated Carbon | 77 K | 2.5 wt% |
| Activated Carbon | 298 K | 1.1 wt% |
| Graphitic Nanofibers | 300 K | 6.2 wt% |
| Al-MOFs | 77 K | 5.1 wt% |
| BN nanoparticles | 291 K | 4.9 wt% |
| SWCNTs | 298 K | 4.5 wt% |
| MWCNTs | 293 K | 4.1 wt% |
| CNTs + Li, K | 300 K | 9.5 wt% |
| CNTs + Pd | 298 K | 8.3 wt% |
| SWCNTs + Ti | 300 K | 8.1 wt% |
| MWCNTs + Pd | 298 K | 0.5 wt% |
| Graphene sheets + Pt | 303 K | 1.6 wt% |

It is worth noting that there are also other ways to store hydrogen at the nanoscale, which show promising success [51], for instance making use of hollow molecular organic frameworks (MOFs) [52]. In all cases where nanostructures keep their stability and morphological integrity, the MB-TDS technique can be applied. Here, we briefly addressed recent progress on adsorbent materials, which can be carbonaceous or non-carbonaceous. The spillover effects of hydrogen molecules on some solid-state adsorbents are prone to achieve highly efficient hydrogen storage, which is an important clue for designing future hydrogen adsorbents.

## 4. Conclusions

The association of plasmas with nanotechnology offers a potential, per se, for significantly increasing the energy efficiency of processes required to mitigate pollution and greenhouse gas emissions (whether using renewable energies as power input or not).

The desirable use of renewable energies in powering plasma reactors helps to ensure full energetic, economic, and environmental sustainability.

Our laboratory equipment allowed the determination of tiny amounts of hydrogen absorbed in a solid medium, which is of great importance in increasing energy efficiency compared to alternative methods of hydrogen storage [48,50]. Also, the development and/or improvement of electrocatalysis or plasma catalysis techniques involves the creation of suitable nanoparticles. With this goal, the decomposition of gaseous mixtures induced with a pulsed laser was ensured in a flow reactor, in which the nanoparticles formed after laser pyrolysis are collected by gravity in a filter. To fully exploit this potential, a fundamental understanding of the effects of plasma on the catalyst and multidisciplinary knowledge of the phenomena that occur in the plasma and at the plasma-catalyst interface is required. In a combined plasma and catalyst procedure, non-thermal plasma reactors that produce reactive species through the emission of photons can start catalytic processes. This may enable operation in a compact space, at room temperature, with or without a catalyst, and with low maintenance costs. Non-thermal plasma reactors generate reactive species through the emission of photons, and these thermal hot spots can initiate catalytic reactions when the plasma is combined with the catalyst. This can allow for operation at room temperature, with or without a catalyst, in a small space, and at a low maintenance cost. Certain carbon nanostructures have the potential to increase hydrogen storage capacity in the solid state, and as carbon-supported catalysts, which together with metallic catalyst nanoparticles, allows for the acceleration of the production of simple fuels. Some other authors have successfully synthesized AgPd alloy nanoparticles loaded on N-ompg-$C_3N_4$ with a considerably high catalytic activity and stability for hydrogen generation [53]. Nanotechnology improvements in using solid-state materials for hydrogen storage is a quite active research area, where recent progress never ceases to amaze us [54].

Future work plans are planned in our laboratory regarding the improvement performances of both electrocatalysts and assembled nanostructures for hydrogen absorption using laser-induced pyrolysis synthesis (where the plasma is expected to give rise to a narrow size distribution of nanoparticles).

In summary, the scientific achievements of nanotechnology supported by nanophysics have the potential to contribute to the transition to a hydrogen-decarbonizing economy by improving the production, storage, and usage of hydrogen. The intention here was to elucidate what is fundamental in this complex subject, encouraging critical thinking, avoiding prioritizing strategic results or even results that have already been tested, and not giving an exhaustive list of hydrogen technologies. The benefits of nanotechnology in the field of hydrogen-based energy transition are presented together with promising perspectives of water electrolysis.

**Funding:** This research was funded (in part) by the Portuguese FCT program, Center of Technology and Systems (CTS) UIDB/00066/2020/UIDP/00066/2020.

**Conflicts of Interest:** The author declare no conflict of interest.

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
