# Peer review of "A Brief on Nano-Based Hydrogen Energy Transition"

_hydrogen, doi:10.3390/hydrogen4030043_

Round 1
Reviewer 1 Report
The manuscript is a review highlighted the most recent aspects of hydrogen production, involving carbon nanotechnology, fuel cells and plasma catalysis. Their features of different processes of hydrogen production were considered. The author suggest that nanotechnology, supported by scientific advances in nanophysics, can contribute to the transition of the economy to green hydrogen by improving the production, storage and use of hydrogen.
Overall opinion: The paper in ready for the publication process
Author Response
Since Reviewer 1 Report is frankly positive regarding all comments and also the English language, I just want to thank for verifying that the manuscript was perfectly understood as well as the future scope of its message.
Reviewer 2 Report
Please try to provide a subheading in methods and processes like the first, hydrogen production methods (by solar-hybrid electrolysis of water, photocatalytic water splitting, and plasma) and divide the hydrogen storage part.
Please try to summarize the production methods in Table for comparison of different methods.
I would like to suggest going through the manuscript again carefully for clarity, syntax and correctness. The English should be improved.
Author Response
The reviewer has shown interest in the manuscript, and he understood the importance of the issues under discussion.
Therefore, I wish to thank his contribution with the suggestions that I am implementing in the new revised version.
I understand the reviewer's desire of including a Table summarizing the hydrogen production methods, but I decided not to do it because the results obtained from the literature are not completely unequivocal regarding the comparison of production efficiencies between them.
Reviewer 3 Report
The manuscript is loosely structured and lacks rigorous logic and narrative structure. The paper needs to focus on the topic and requires an in-depth technical analysis of the application of nanotechnology in hydrogen energy transition. This manuscript needs to be modified according to the following comments:
1)The novelty and originality of this manuscript should be emphasized in the abstract or the introduction. Besides, there is needed to add a paragraph at the end of the introduction describing what will be discussed in this manuscript.
2)The abstract does not reflect the main achieved results. It should be modified.
3)The paper's novelty is weak and should be highlighted, and the author should emphasize the work done by yourself, which is different from the work done by existing researchers.
4)The title and contents of the paper lack a connection: for example, "Green Hydrogen Benefits" does not relate to the topic of "nanotechnology" directly, and the contents under the title "Green Hydrogen Benefits" do not discuss the advantages of Green Hydrogen, such as reducing carbon emissions and promoting industrial transformation. The section on "Methods and Processes" does not include the methods and processes of the scientific experiments.
5) The conclusion is not well. Also, future work plans should be included in the conclusion.
6) Since the article is a compilation, few papers have been reviewed, or opinions have been supported. More references and information should be added.
Some spelling and other grammatical errors should be modified. Language issues should be double checked again carefully reading the whole manuscript.
Author Response
The author wishes to thank the reviewer for his comments and positive suggestions.
Structure has been improved for ensuring a logic narrative. Worth to note that this a brief narrative highlighting critical and pertinent issues in hydrogen technologies that can benefit from recent advances in nanotechnology and not a standard review manuscript.
This manuscript addresses in a concise and original way the importance of include nanotechnology in both green electroproduction of hydrogen and hydrogen storage in solid media.
After stressing the importance and benefits of green hydrogen production, this work will emphasize the main methods and processes of hydrogen production and storage involving nanotechnology, with a special focus in what is under experimental research in the author’s laboratory regarding these two subjects.
The abstract was modified accordingly.
The paper´s novelty is included in the text as well as in the references.
At first sight green hydrogen benefits does not relate directly to nanotechnology but let us have a deeper look into the subject. It is never too much address that presently, for reducing carbon emissions and promoting an industrial sustainable transformation, the green hydrogen is indispensable. In addition, the electroproduction of hydrogen using renewable energy is an effective way of rising the energy efficiency since it can work independently of the basic electric grid. Thus, this constitutes per se a first contribution to the increase of hydrogen production efficiency. Furthermore, when electroproduction performance is enhanced by catalyst effects acting at the nanoscale, one can add a second contribution to hydrogen production efficiency increase.
Future work plans are planned in our laboratory regarding the improvement performances of both electrocatalysts and assembled nanostructures for hydrogen absorption, using laser induced pyrolysis synthesis (where the plasma is expected to give rise to narrow size distribution of nanoparticles).
Indeed, one can consider the article as a brief compilation (not an extended review) and so only some relevant references were considered; anyway a few more recent significant references were included.
Extensive editing of English language was done.
Reviewer 4 Report
In the manuscript " A Brief on Nano-based Hydrogen Energy Transition" Rui F. M. Lobo et al. intends to change eventual beliefs that hydrogen technologies are being imposed only for reasons of sustainability and not for the intrinsic value of the technology itself. This manuscript is well-organized and carefully written. It can be accepted after minor revision. The comments are presented as follows:
1. The latest literature about hydrogen storage technologies should be cited, such as Metal organic framework supported niobium pentoxide nanoparticles with exceptional catalytic effect on hydrogen storage behavior of MgH2. L. Zhang, F.M. Nyahuma, H. Zhang, C. Cheng, J. Zheng, F. Wu, L. Chen. Green Energy and Environment, 2022, https://doi.org/10.1016/j.gee.2021.09.004. Chao Wan, Liu Zhou, Suman Xu, Biyu Jin, Xin Ge, Xing Qian, Lixin Xu, Fengqiu Chen, Xiaoli Zhan, Yongrong Yang, Dangguo Cheng. Defect engineered mesoporous graphitic carbon nitride modified with AgPd nanoparticles for enhanced photocatalytic hydrogen evolution from formic acid, Chemical Engineering Journal, 2022, 429, 132388. Wang Yaxiong, Zhong Shunbin, Sun Fengchun. Research Progress in Vehicular High Mass Density Solid Hydrogen Storage Materials. Chinese Journal of Rare Metals, 2022, 46, 796-812.
2. What is the main question addressed by the research?
3. What does it add to the subject area compared with other published material?
4. Do you consider the topic original or relevant in the field? Does itaddress a specific gap in the field?
Author Response
Thanking the reviewer who have shown interest in the article and in the understanding of the importance of the issues under discussion, I just come to clarify the comments he made:
1-The suggested references have been considered, by including them in the revised text.
2-This research mainly addresses the importance of incorporating nanotechnology, either in terms of innovative methods of hydrogen storage or production methods.
3-Compared to other published material it adds the intrinsic value of new processes based on nanotechnology for electrocatalytic hydrogen production associated with appropriate safe storage techniques in solid state.
4-The topic is relevant in the field of hydrogen technology, by filling the current mind gap in believing that present processes are mature and enough developed to ensure the so desired energy sustainability. On the contrary, nano-engineering based on nanophysics have the potential to significantly change the paradigm of conventional hydrogen technologies.
Round 2
Reviewer 3 Report
1. Reviewed Submission
1.1. Recommendation
Major Review
2. Comments to Author:
No.: Hydrogen-2516353
Title: A Brief on Nano-based Hydrogen Energy Transition
Overview and general recommendation:
The manuscript entitled “A Brief on Nano-based Hydrogen Energy Transition” covers the journal subject. The revised version has added novelties and future technological trends to the paper, improved the references, and revised grammatical errors. However, the paper still needs to be refined to meet the requirements for publication:
1. Some titles are too short to see the connections with the contents below. For example, the title “Hydrogen Production”, “Plasma” and “Hydrogen Storage” should be modified.
2. There are explicit spelling and other grammatical errors which should be checked again carefully. For example, “At first sight green hydrogen benefits do not relate directly to nanotechnology but let us have a deeper look into the subject.”
3. If the authors have time, please also arrange the logical structure and main content of the paper by the requirements of a compilation article. An example of this would be to follow the sequence of the hydrogen energy industry chain, from renewable energy production, hydrogen production, hydrogen storage, and hydrogen transmission, to end-use, with a combination of the various links and the latest developments in nanotechnology.
Language issues (spelling and grammar) should be double-checked again carefully reading the whole manuscript.
Author Response
The author thanks the reviewer for the additional comments and made corrections accordingly, whenever possible.
1) The titles mentioned by the reviewer were slightly lengthened (blue color) in a way that it becomes clearer to see the connections with the contents.
2) The spelling and the grammar were checked again, and the corrections are highlighted in blue.
3) As far as possible and assuming that it would always be possible to improve, the author made some changes (highlighted in blue) under penalty of not being able to change the structure, as this would imply lengthening the text too much, which would go against the purpose of the title ("A Brief on…").
Anyway, those specific changes were made seeking to follow the sequence of the hydrogen energy industry chain, from renewable energy production, hydrogen production, hydrogen storage, and hydrogen transmission, to end-use, with a combination of the various links and the latest developments in nanotechnology.